# Clinical Characteristics, Diagnostic Approach and Outcome of Thyroid Incidental Findings vs. Clinically Overt Thyroid Nodules: An Observational Single-Centre Study

**DOI:** 10.3390/cancers15082350

**Published:** 2023-04-18

**Authors:** Tom Jansen, Nike Stikkelbroeck, Annenienke van de Ven, Ilse van Engen-van Grunsven, Marcel Janssen, Han Bonenkamp, Martin Gotthardt, Romana T. Netea-Maier

**Affiliations:** 1Department of Internal Medicine, Division of Endocrinology, Radboud University Medical Center, Geert Groteplein Zuid 10, 6525 GA Nijmegen, The Netherlands; 2Department of Pathology, Radboud University Medical Center, Geert Groteplein Zuid 10, 6525 GA Nijmegen, The Netherlands; 3Department of Radiology and Nuclear Medicine, Radboud University Medical Center, Geert Groteplein Zuid 10, 6525 GA Nijmegen, The Netherlands; 4Department of Surgery, Radboud University Medical Center, Geert Groteplein Zuid 10, 6525 GA Nijmegen, The Netherlands

**Keywords:** thyroid nodules, incidentaloma

## Abstract

**Simple Summary:**

Thyroid nodules are common and can present as visible, palpable or symptomatic nodules (non-incidentalomas) and as coincidental findings on imaging techniques (so-called incidentalomas). The majority are benign but recognizing clinically relevant nodules remains a challenge. Dutch guidelines currently recommend to refrain from additional diagnostic testing in incidentalomas other than FDG-PET-incidentalomas. However, there is no consensus on, or data of, the outcome of the further approach. Our retrospective observational study aims to compare clinical characteristics and outcome between patients with incidentalomas and non-incidentalomas. We found that the risk of malignancy in incidentalomas found on other modalities than FDG-PET was significantly lower (2.8%) than FDG-PET-incidentalomas (11.8%) or non-incidentalomas (11.1%). Furthermore, incidentalomas were significantly smaller than non-incidentalomas. Our findings support the current recommendations to prioritize additional analysis to non-incidentalomas, FDG-PET incidentalomas and clinically relevant non-PET-incidentalomas. These findings are relevant to avoid unnecessary diagnostic testing and therapy and therefore possible harm to patients.

**Abstract:**

**Context**: Thyroid nodules are common and can present as clinically overt nodules (visible, palpable or symptomatic nodules) and so-called incidentalomas (coincidental findings on imaging techniques). The majority are benign but recognizing clinically relevant nodules remains a challenge. Current Dutch guidelines recommend to refrain from additional diagnostic testing in incidentalomas other than FDG-PET-incidentalomas, unless there are suspicious clinical and/or sonographic features. However, there is no consensus on the further approach and no “real-life” data on the outcome of such an approach. **Objective**: To compare clinical characteristics, diagnostic approaches and clinical outcome between patients referred with thyroid incidentalomas and non-incidentalomas at one academic referral thyroid clinic. **Methods**: Clinical and demographical characteristics, diagnostic and therapeutic approaches and outcome were retrospectively obtained from the files of all patients newly referred because of thyroid incidentalomas or non-incidentalomas to our institution (between March 2011 and January 2017). Subsequently, the data were compared between both groups. **Results**: In total, 351 patients (64.3%) were referred because of non-incidentalomas and 195 (35.7%) because of incidentalomas. Incidentalomas were smaller (48.7% <2 cm) than non-incidentalomas (23.4% <2 cm). Furthermore, incidentalomas were less often symptomatic (15.9 vs. 42.7% *p* < 0.001). Fine-needle aspiration was performed in a similar percentage of the patients in the two groups (62.6% of incidentalomas vs. 69.8% in non-incidentaloma, *p* = 0.08). Significantly less malignancies were found among incidentalomas compared to non-incidentalomas (5.1% vs. 11.1%, *p* = 0.019). Moreover, significantly more malignancies occurred in PET-incidentalomas than non-PET-incidentalomas (11.8% vs. 2.8%, *p* = 0.023). In fact, the proportion of malignancies in PET-incidentalomas and non-incidentalomas was similar (11.8% vs. 11.1%, *p* = 0.895). Stability or decrease in size was observed in 96.5% of nodules receiving ultrasound follow-up. **Conclusions**: Patients with small asymptomatic thyroid incidentalomas represent an important proportion of the patients referred for additional diagnostic evaluation. The risk of malignancy in these patients is lower than in those with symptomatic palpable lesions, particularly in the patients with incidentalomas discovered on CT, MRI or US. Our findings support the current recommendations from the Dutch guidelines to not indiscriminately perform additional analysis and treatment on all incidentalomas, but prioritize this to FDG-PET-incidentalomas and clinically relevant non-PET-incidentalomas. Moreover, US features can further refine the selection of the patients who require immediate FNAC and/or surgery.

## 1. Introduction

Thyroid nodules are highly prevalent with approximately 5% of the adult population having a palpable nodule [1]. Thyroid ultrasound (US) can detect nodules in up to 68% of individuals [1]. Once a nodule is discovered, the primary goal of additional diagnostic examinations and follow up is to exclude thyroid malignancies. In contrast to the high incidence of thyroid nodules, thyroid carcinoma (TC) is relatively rare [2,3]. Nevertheless, the incidence of TC, particularly of small (<2 cm) often clinically indolent papillary TC, has increased significantly worldwide over the last decennia, most likely as a consequence of a raise in imaging rates and diagnostic scrutiny which have made incidental thyroid nodules a more common finding [4]. Distinguishing between clinically relevant and indolent nodules remains a challenge in the clinical practice [5].

Thyroid nodules are typically found either at physical examination (non-incidentaloma) or incidentally when imagining studies are performed for other indications than the thyroid gland (incidentaloma). It is estimated that around 5–10% of non-incidentalomas are malignant [6,7,8,9]. Therefore, the majority of guidelines, including the Dutch guideline on TC recommend performing additional diagnostic workup to exclude malignancy in palpable and symptomatic thyroid nodules [1,10,11]. 

Incidentalomas are mostly discovered on neck US or CT-scans and less often on magnetic resonance imaging (MRI) or ^18^F-fluorodeoxyglucose-positron emission tomography (FDG-PET) scans [12]. Generally, they are smaller than 2 cm [13,14,15,16]. The risk for malignancy in incidentalomas is presumed to be lower than that of palpable nodules [13,17]. This risk can be higher in FDG-PET-discovered incidentalomas [18], though many studies are ambiguous and are affected by selection bias [12]. Moreover, no clear consensus has been reached about the optimal assessment and therapeutic management of thyroid incidentalomas [12,17,19]. The most recent guideline of the American Thyroid Association does not distinguish between the approach towards incidentaloma vs. non-incidentaloma with the exception of FDG-PET incidentaloma [1]. The Dutch guideline on TC recommends to refrain from routinely performing further diagnostic tests in case of incidentalomas other than FDG-PET-incidentalomas, unless the lesion is palpable on clinical examination, there are clinical and/or sonographic features that can raise the suspicion of malignancy and/or the patient needs reassuring [10]. However, no clear protocol on further follow-up is available. Consequently, historically, the patients with thyroid nodules are not routinely recommended regular follow-up unless they are symptomatic or at patients’s request. Furthermore, there is no data available on diagnostic or therapeutic approach routinely followed by Dutch physicians nor on the outcome of this approach. Lastly, we could find only one previous study that compared clinical features and outcome of patients with non-incidentally found nodules to those with US-incidentalomas in the same clinical practice environment [20]. 

Therefore, the aim of this study is to compare the clinical characteristics, diagnostic approaches and clinical outcome between patients presenting with thyroid incidentalomas and patients with non-incidentalomas, referred to a tertiary university hospital in the Netherlands. The results will give more insight into the features distinguishing these types of nodules and into the diagnostic and therapeutic strategies followed by Dutch physicians in a “real-life” practice.

## 2. Materials and Methods

### 2.1. Study Population and Design

The study was performed in accordance with the requirements of the institutional medical ethical committee (Research Ethics Committee of the Radboud University Medical Center file number 2023-16181). We conducted a retrospective cohort study. All patients with thyroid nodules who were newly referred to the thyroid nodule outpatient clinic of the Radboud University Medical Center, Nijmegen, between March 2011 and January 2017 were included. Subsequently, data was collected and analyzed in 2018. The patients who were referred to the outpatient clinic for any other reason than thyroid nodules (including the patients with known TC referred for second opinions and patients referred specifically for known thyroid dysfunction) and the patients who came for the follow-up of (nodular) goiter or thyroid nodules were excluded. 

### 2.2. Data Collection

Clinical and demographical characteristics, diagnostic and therapeutic approach, and outcome were retrospectively obtained from the electronic patient files of all included patients. 

### 2.3. The Workflow of the Thyroid Nodule Clinic

After history taking, all patients underwent physical examination, thyroid function was assessed (in the patients in which euthyroidism was confirmed at referral by either the family practitioner or the referring specialist, the thyroid function measurement was not repeated) and a hands-on US, performed by the examining endocrinologist. All thyroid US were performed by the same experienced three operators (RN-M, NS and AvdV) on the same machine. At the time of the diagnosis the US assessment according to Thyroid Imaging Reporting and Data System (TIRADS) [21] classification was not routinely implemented, but the separate US characteristics of the nodules were registered. The endocrinologist discussed the findings with the patient and the indication for a fine needle aspiration cytology (FNAC) as well as the advantages, disadvantages and consequences of additional tests in a shared-decision manner. An indication for a FNAC was considered in the presence of suspicious clinical and US features [1] or in the presence of other factors associated with an increased risk of malignancy such as a positive family history of TC or previous ionized radiation exposure; or when patients expressed their explicit wish to have a FNAC for reassurance even in the absence of suspicious findings. Additionally, in case of FDG-uptake in the nodule on the FDG-PET scan, a FNAC was considered in nodules larger than 1 cm, but not in nodules smaller than 1 cm without suspicious US findings or other risk factors unless the patient explicitly required this for reassuring. Because our center is a tertiary referral center, many patients had FNAC performed elsewhere, before the referral. If a FNAC had been performed before referral, this FNAC was revised in most of the cases and the indication for repeating the FNAC was critically assessed. The FNAC was examined on site to ensure sufficient sample. In case of two insufficient passes, the FNAC was repeated the next day in most of the patients. Cytological results were reported according to the Bethesda classification system [22]. 

Based on the final result of the FNAC different approaches were recommended: (1) no follow-up in case of small, asymptomatic, cytologically benign nodules, unless symptoms or nodule growth occurred; (2) follow-up at our clinic or at the referring center elsewhere when patients were referred form another center, (3) surgery, either lobectomy or total thyroidectomy in case of either clinical or suspicious US characteristics regardless the FNAC results, inconclusive, suspicious or malignant FNAC or presence of (obstructive) complaints. The decision to recommend follow-up was at the liberty of the physician after discussing this with the patient, as no routine follow-up is recommended according to the Dutch national guidelines. For the patients who were not recommended follow-up, they and their family practitioner were recommended to contact the specialist when they developed mechanical or cosmetic complaints and/or there was a suspicion of nodule growth. For the patients who were referred for follow-up elsewhere, this took place in one of the regional hospitals affiliated to our center and the referring physician was requested to contact our center when there was progression. The patients in which a malignant thyroid tumor was detected during the follow-up, were discussed multidisciplinary in our regional tumor board. 

### 2.4. Data-Analysis

The study population was divided into two cohorts: patients primarily referred for incidentalomas and patients primarily referred for non-incidentalomas. The latter cohort included patients referred for visible, palpable or symptomatic nodules (e.g., obstructive complaints, hoarseness, pain) or goiter. Next, these two cohorts were compared on the basis of the above-mentioned variables. An independent T-test (two-sided) was used to determine differences between continuous variables (e.g., age). Most variables were of the categorical type, e.g., sex or Bethesda-classification, where X^2^-tests or Fisher Exact Tests were used to identify differences. Where informative, 95% confidence intervals (95%-CI) were calculated. *p*-values below 0.05 were considered as statistically significant. Statistical Software Package (SPSS, version 22.0, IBM Corp., Somers, NY, USA) was used to build our database and statistically analyze the data.

## 3. Results

### 3.1. Clinical and Demographical Characteristics

The patients flow throughout the study is presented schematically in Figure 1. In total, 546 patients were included. The clinical characteristics of these patients are reported in Table 1. Of the 546 patients, 195 (35.7%) were referred because of incidentalomas and 351 because of non-incidentalomas. Of the incidentalomas, 51/195 (26.2%) were discovered on FDG-PET-scans, 72/195 (36.9%) on neck US, 43/195 (22.1%) on CT-scans, 28/195 (14.4%) on MRI-scans and 1/195 (0.5%) on thorax radiography. Of the patients with non-incidentalomas, 12/351 (3.4%) were referred because of a clinically suspected TC. Clinically, the incidentalomas were significantly smaller, less often palpable on physical examination and were less likely to cause nodule-related obstructive symptoms. Slightly less patients with incidentaloma had thyroid function tests indicating thyroid dysfunction. None of the patients with thyroid dysfunction had symptoms related to this and the thyroid dysfunction was not known before referral.

### 3.2. Outcomes of FNAC

In total 128 nodules in 122/195 (62.6%) patients with incidentalomas, and 257 nodules in 245/351 (69.8%) patients with non-incidentalomas, had a FNAC. One patient who was planned to have the FNAC after bridging the anticoagulant therapy died of an unrelated cause before the FNAC could take place. 

The outcome of FNAC in which the highest Bethesda classification was considered in case of repeated FNAC, is depicted in Figure 2. 

### 3.3. Follow-Up Outcome

Significantly more patients with incidentalomas (65/195 (33.3%)) than non-incidentalomas (85/351 (24.2%)) received no follow-up recommendation (*p* = 0.022). In the remaining patients either some form of follow-up or surgery was recommended (Figure 1). 

Of the 262 patients who were recommended follow-up, 202 (75/195 incidentalomas and 127/351 non-incidentalomas) received follow-up at our clinic with repeated US. The majority of nodules remained stable or decreased in size. Only 7 nodules showed limited growth between 10–20% (3 non-incidentalomas, 4 FDG-PET incidentalomas). FNAC was performed in 6 of these growing nodules, showing Bethesda classification 2 in 5 nodules and Bethesda 1 only in one. Four of the 6 patients with growing nodules had subsequent surgery, which revealed a benign histology in all, and two patients remained in the follow-up.

In total 60 patients (25/195 (12.8%) incidentalomas and 35/351 (10.0%) non-incidentalomas) either had follow-up conducted by their primary care provider or had their follow-up at a different clinic and there was no documentation available concerning their follow-up. They are referred as lost to follow-up in Figure 1. This was not significantly different between the two groups. Of the 202 patients who were followed up, 14 (6.9%) received other treatments (e.g., radioactive iodine, thyrostatic medication). 

Of the FDG-PET incidentalomas, 27/51 (52.9%) were recommended follow-up, which in 22/51 (43.1%) patients took place at our clinic, showing stable size in 21/22 and a decrease in size in 1/22. Eleven (11/51 (21.6%)) patients had an indication for surgery, which was not significantly different compared to non-PET-incidentalomas (19/144 (13.2%)) (*p* = 0.154). Two patients with Bethesda 6 cytology did not undergo surgery because of comorbidity (Figure 1)

### 3.4. Surgery Outcome

Surgery was recommended to 134/546 patients (24.5%), significantly more often for non-incidentalomas (104/351 (29.6%)) than incidentalomas (30/195 (15.4%)) (*p* < 0.001). Histological results were available for 128 patients and lacked in the remaining six patients (did not undergo surgery (n = 3), lost to follow-up (n = 2) or surgery elsewhere and results not available (n = 1)). Based on histological and cytological (for the six patients who did not undergo surgery despite malignant cytology) results, 39/351 (11.1%) non-incidentalomas were malignant versus 10/195 (5.1%) incidentalomas. Among those patients who had surgery recommended, this is not significantly different (39/104 (37.5%) non-incidentaloma vs. 10/30 incidentaloma (33.3%), *p* = 0.651) (Figure 3). However, within the whole cohort, the prevalence of TC was significantly higher in the non-incidentaloma than in the incidentaloma group (11.1% vs. 5.1%, *p* = 0.019) (Figure 4). Moreover, significantly more malignancies were found in patients with an FDG-PET-incidentaloma (6/51 (11.8%) than in those with non-FDG-PET-incidentalomas (4/144 (2.8%)) (*p* = 0.023). The prevalence of malignancy among the FDG-PET incidentalomas was however not different than that among the non-incidentalomas. Furthermore, non-PET-incidentalomas (4/144 (2.8%) contained significantly fewer malignancies than non-incidentalomas (39/351 (11.1%)) (*p* = 0.003. The proportion of malignancies in patients with insufficient cytology (Bethesda 1) was 3% (n = 2). 

The clinical characteristics and the outcome of the patients with malignancies are presented in Table 2. Among them, five were patients with an anaplastic TC, most likely a selection bias related to the fact that our center is a tertiary referral center for patients with a suspicion of aggressive thyroid malignancy. The five patients in which no surgery was performed but had malignancy proven by cytology, the malignancies included: one papillary TC, two medullary TC, one anaplastic TC and one metastasis from renal cell carcinoma. 

## 4. Discussion

Worldwide, it is nowadays recognized that many patients with thyroid cancer have indolent tumors and while diagnosing and treatment of these tumors may not significantly impact on the life expectancy of the patients it can elicit important burden for both the individual patients and the society. For this reason, the focus has clearly shifted worldwide towards identification of patients who are most likely to benefit from an early diagnosis and treatment rather than identifying clinically indolent cases. In this study we compared the clinical presentation, diagnostic and therapeutic approaches, and outcome in patients referred for thyroid incidentalomas and non-incidentalomas in the context of a “real-life” referral thyroid clinic in The Netherlands, where a restrictive diagnostic approach is recommended according to the national guidelines published in 2015. We found that thyroid incidentalomas represent a large percentage of the referred patients and a majority of them had smaller and non-symptomatic lesions compared with thyroid non-incidentalomas. In the setting of the outpatient clinic, incidentalomas and non-incidentalomas were both approached in a similar and thorough manner, considering the clinical context of the patients, their concerns and preferences and the clinical and sonographic features of the nodules. This approach has resulted in comparable proportions of patients undergoing additional investigations by FNAC. Despite this, the prevalence of malignancies among incidentalomas was significantly lower than among non-incidentalomas. Furthermore, within the incidentaloma group there was a significant difference between the FDG-PET-incidentalomas and the non-PET-incidentalomas with the latter showing the lowest proportion of malignant tumors. This supports the current recommendations to prioritize additional investigations in FDG-PET-incidentaloma and clinically relevant non-PET incidentaloma.

The prevalence of malignancies among non-incidentalomas of 11.0% in our series falls within the high range of the estimated 5–14% based on previous literature [6,7,8,9,23], probably as result of referral bias to our tertiary referral center. The risk of malignancy among incidentalomas has been reported to be higher in FDG-PET incidentalomas than in non-PET incidentalomas (the latter including CT, MRI and US-discovered incidentalomas) [14,17,18,24,25,26,27]. Several large retrospective studies have reported on the prevalence of TC among non-PET incidentalomas, with prevalence rates ranging between 2.3 and 11.3% in CT-incidentalomas, 8% in MRI-incidentalomas and approximately 5–6% in US-incidentalomas [14,17,26]. The prevalence of malignancy among non-PET incidentalomas in our series falls within the low range of these previously reported results. However, higher malignancy rates have been reported in FDG-PET-incidentalomas, with two large reviews reporting malignancy rates of 34.8 and 34.6%, respectively [24,27]. Flukes and colleagues reported 39.6%, whereas a smaller rate of 19.8% was reported in a recent meta-analysis [18,25]. In our series, only 11.8% of PET-incidentalomas were malignant, which is comparable with the 16.6% rate reported by Thuillier et al. [28] and the 10.9% rated reported by Makis et al. [29]. One possible explanation for this relatively low rate is that we did not utilize the maximum standardized uptake values (SUV_max_) to select nodules for additional investigations, as SUV_max_ has insufficient power to accurately discriminate between benign and malignant or suspicious nodules. Furthermore, we have not reviewed the FDG-PET-scans to confirm that the mentioned FDG-uptake was focal, therefore a few patients with more diffuse FDG-uptake might have been included as well. In the latter, much lower incidence of malignancy of 4.4% has been reported [27]. Additionally, some patients did not undergo FNAC due to lack of clinical consequences related to severe comorbidity or technical issues (e.g., nodule could not be properly reached), which might have led to missing a few clinically irrelevant malignancies. Nonetheless, FNAC was performed in 82.4% of all PET-incidentalomas in our series whereas in other studies on PET-incidentalomas FNAC has been performed in only 50–60% of the included patients [28,29]. On the other hand, selection of the patients with the highest probability of malignancy to undergo histological confirmation might have led to an overestimation of the presumed overall risk as reported in the literature [27]. The large heterogeneity in study populations, selection bias and employed methodologies makes comparison between the results of different studies difficult. 

When approaching incidentalomas and particularly PET-incidentalomas, it is important to also take into account the clinical context of the patients, instead of indiscriminately performing additional diagnostics when there is no evidence of a clinical benefit [30]. Our analysis showed that of all the nodules that had US follow-up, only 3.0% showed growth over time and in FDG-PET-incidentalomas specifically (n = 22), none showed growth, making a clinically relevant malignancy unlikely. Moreover, the vast majority of FNAC-proven benign thyroid nodules show no growth over time [31]. Therefore, US follow-up can be a good alternative for patients without suspect features and/or small nodules. 

FNAC performed on incidentalomas yielded more insufficient results (Bethesda 1) than in non-incidentalomas which is in accordance with other studies [28]. This may be related to the location of these nodules, making them less easy to approach, resulting in FNAC causing more discomfort for patients and subsequently, less optimal procedures and more reluctance to repeat the FNAC. Less patients with an insufficient FNAC in the incidentaloma group received a second FNAC than in the non-incidentaloma group. Additionally, the physician performing the FNAC may have been biased in the decision to repeat this by knowledge from the previous literature of the small estimated risk of malignancy in case of (small) incidentalomas, a decision also supported by the recommendations of the Dutch guidelines. Furthermore, the smaller size of these nodules may also contribute to the higher rate of insufficient sample, since previous research has shown that a smaller size of thyroid nodules is associated with a higher chance of inadequate cytology [32,33]. 

Because of the small number of patients in our incidentaloma group it was not possible to investigate potential factors associated with malignancy in our series. As expected, patients with incidentalomas had significantly less complaints and less palpable lesions. Furthermore, half of the nodules was smaller than 2 cm which is not surprising and in accordance with previous literature [14,15,16,34]. Nonetheless, size alone is not sufficient to discriminate between benign and malignant thyroid lesions [35,36]. Clinical or sonographic characteristics can be used to estimate the risk of a malignancy but are unable to reliably identify malignant nodules. Furthermore, they cannot accurately distinguish between aggressive nodules which require treatment and indolent ones that do not [37]. Therefore, with respect to incidentalomas, the risk of over-diagnosis and overtreatment remains a major concern leading in some countries to a dramatic rise in incidence of (mainly papillary) TC, many of which are likely subclinical and nonlethal [4,5]. Given the large number of discovered incidentalomas this likely affects the costs of medical care and causes unnecessary burden for patients. 

Our study has some limitations. First of all, it is retrospective and our series is relatively small. Nonetheless, the number of patients with PET-incidentalomas included is comparable with many previous studies [24,27]. Secondly, our series came from a single, tertiary referral institution. Consequently, our results may not be entirely generalizable to the rest of the population with thyroid nodules. Thirdly, not all patients underwent surgery and therefore some malignancies might have been missed, despite of the absence of suspect features and no progression during follow-up. If this was the case, they likely were low-risk TC’s. For example, Moon et al. [34] very recently published a retrospective study that showed that thyroid malignancies discovered by screening were significantly smaller, had a less advanced T stage and had a significantly lower all-cause and thyroid cancer related mortality. In absence of high risk features, these low risk tumors (mainly papillary microcarcinomas (PMC)), have an excellent oncological outcome [38]. 

In our series, in 10.9% of the patients who were recommended follow-up, no documentation was available on their outcome. However, both the primary care physician and the specialists elsewhere to which the follow-up was entrusted, were requested to report if clinically relevant progression of malignancy was diagnosed during follow-up. To date, this was not the case. Relatively short follow-up may have obscured slowly progressing TC. Lastly, we cannot reassess the US features according to the TIRADS classification because this was at the time not routinely implemented in our country. 

## 5. Conclusions

In conclusion, patients with small asymptomatic thyroid nodules discovered incidentally represent an important proportion of the patients referred for additional diagnostic evaluation. Our results show that, while undergoing a similar diagnostic approach, the risk of malignancy in non-PET incidentalomas is lower than in those with symptomatic palpable lesions. Furthermore, our results confirm that the risk of malignancy is higher in FDG-PET thyroid incidentalomas. The ATA recommendations to determine the indication for FNAC based on the sonographic characteristics of the nodules regardless the detection route has already been proven useful in reducing the number of unnecessary diagnostic procedures. Nonetheless, thorough examination, clinical reasoning and when indicated additional diagnostic procedures such as FNAC ensure identifying the patients with increased risk of clinically relevant malignancy and reassurance of those who may not require treatment. These findings support the current recommendations of the Dutch guideline to not indiscriminately perform additional analysis and treatment of all incidentalomas, but prioritize on PET-incidentalomas and clinically relevant non-PET-incidentalomas. Moreover, US features can help further refining the selection of the patients who require immediate FNAC and/or surgery instead of reassuring or follow-up. 

## Figures and Tables

**Figure 1 cancers-15-02350-f001:**
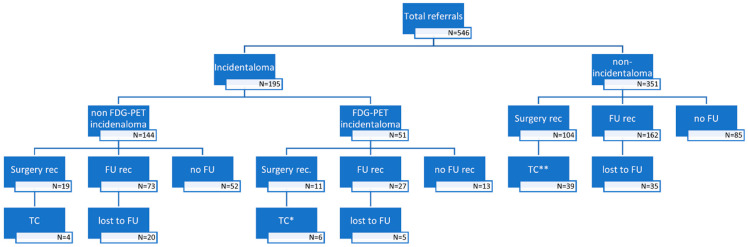
Patients flow and outcome. FU: follow-up; Surgery rec: surgery recommended; FU rec: follow-up recommended; no FU rec: no follow-up recommended; TC: thyroid carcinoma; * including 2 patients who did not undergo surgery but cytology indicated malignancy; ** including 4 patients who did not undergo surgery but cytology indicated malignancy.

**Figure 2 cancers-15-02350-f002:**
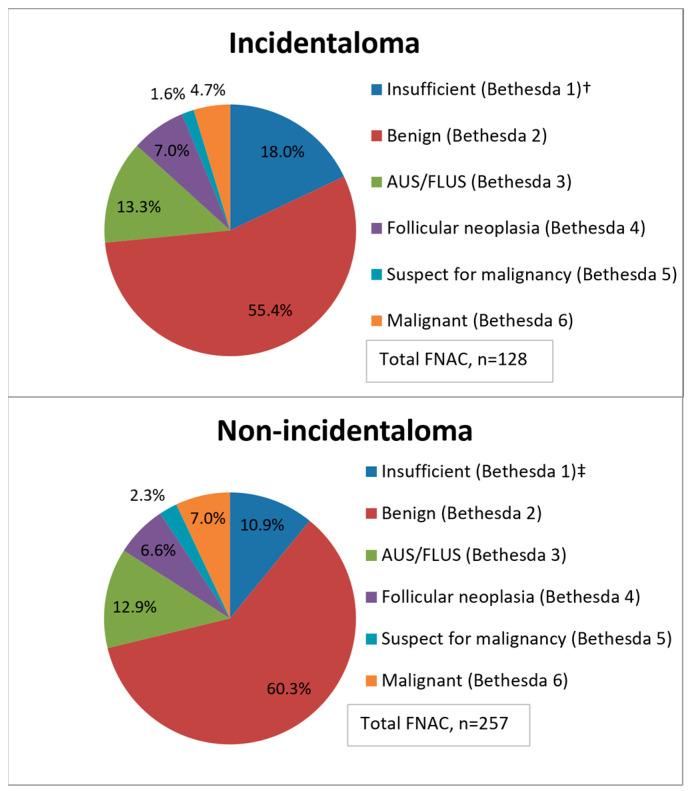
Outcome of FNAC. In case of repeated FNA the highest Bethesda result was considered. The 12 purely cystic nodules in which FNAC was performed for the main purpose of drainage were not included in this analysis. † not including 1 FNAC performed on a cystic lesion. ‡ not including 11 FNAC performed on cystic lesions.

**Figure 3 cancers-15-02350-f003:**
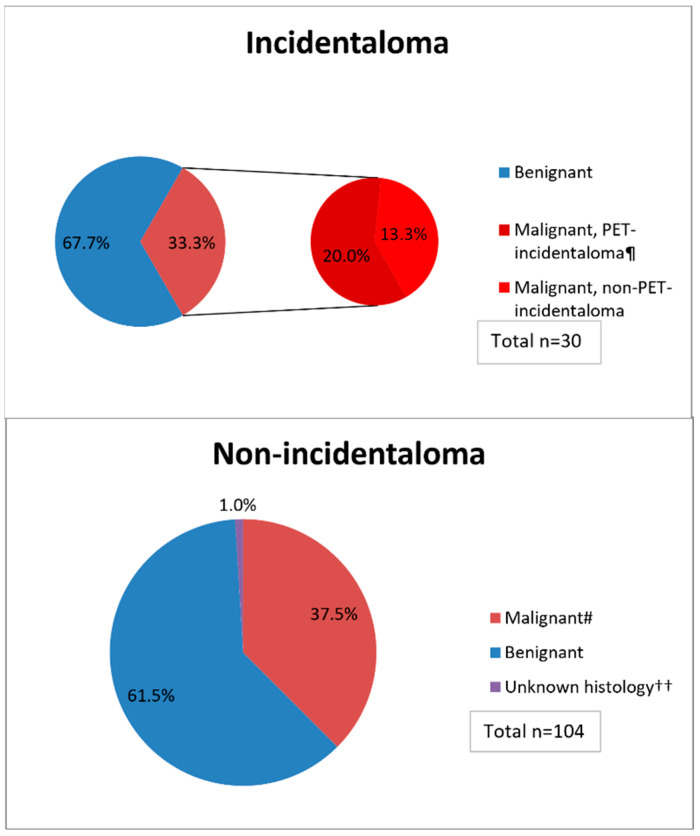
Proportion of malignant tumors within the patients who had surgery recommended. “¶” including 2 patients who had proven malignancy but did not receive surgery. # including 3 patients who had proven malignancy but did not receive surgery. †† including 1 patient who received surgery in a different hospital and 1 who refused surgery.

**Figure 4 cancers-15-02350-f004:**
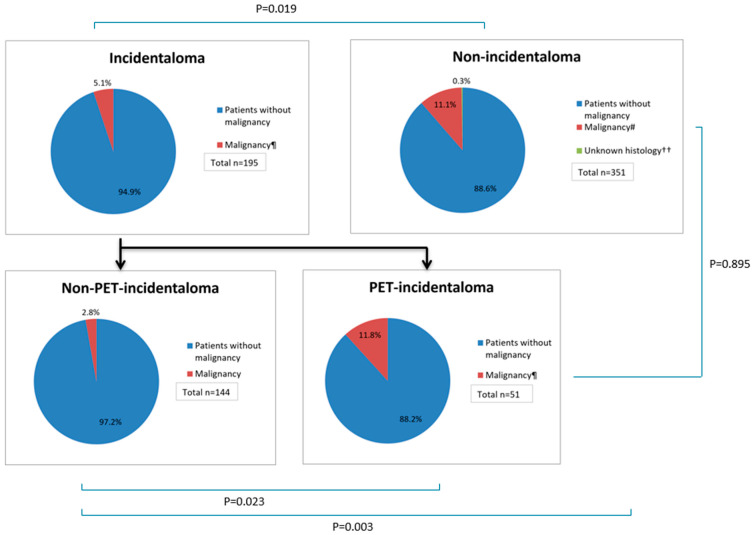
Proportion of malignant tumors among the whole study population. “¶” including 2 patients who had proven malignancy but did not receive surgery. # including 3 patients who had proven malignancy but did not receive surgery. †† including 2 patients who received surgery in a different hospital and 1 who refused surgery.

**Table 1 cancers-15-02350-t001:** Clinical characteristics of the patients.

	Incidentalomas (n = 195)	Non-Incidentalomas (n = 351)	Total (n = 546)	*p*-Value ^#^
Female sex, n (%)	132 (67.7)	273 (77.8)	405 (74.2)	0.010
Mean age ± SD (years)	58.7 ± 14.5	50.6 ± 14.1	53.5 ± 14.8	<0.001
Median duration of follow-up (range) (years)	2.93 (0.9–6.7)	2.91 (0.9–6.7)	2.92(0.9–6.7)	0.585
Referring physician:				
*Family physician, n* (%)	36 (18.5)	193 (55.0)	229 (41.9)	<0.001
*Specialist, n* (%)	159 (81.5)	158 (45.0)	317 (58.1)	<0.001
**Symptoms**				
Visible/palpable nodule, n (%)	21 (10.8)	268 (76.4)	289 (52.9)	<0.001
Obstructive complaints, n (%)	31 (15.9)	150 (42.7)	181 (33.2)	<0.001
**Risk factors for malignancy**				
Previous radiotherapy, n (%)	4 (2.1)	13 (3.7)	17 (3.1)	0.273
Family history of thyroid malignancy, n (%)	11 (5.6)	21 (6.0)	32 (5.9)	0.868
**Physical examination**				
Palpable lesion				
*Palpable (dominant) nodule, n* (%)	68 (34.9)	267 (76.1)	335 (61.4)	<0.001
*Diffuse goiter, n* (%)	3 (1.5)	23 (6.6)	26 (4.8)	0.008
No palpable lesions, n (%)	124 (63.3)	61 (17.4)	185 (33.9)	<0.001
**Thyroid function tests**				
Euthyroidism, n (%)	168 (86.2)	287 (81.8)	455 (83.3) *	0.038
Hypothyroidism, n (%)	6 (3.1)	29 (8.3)	35 (6.4)	0.020
*Of which subclinical, n* (%)	5 (2.6)	15 (4.3)	20 (3.7)	0.154
Hyperthyroidism, n (%)	8 (4.1)	17 (4.8)	25 (4.6)	0.720
*Of which subclinical, n* (%)	6 (3.1)	9 (2.6)	15 (2.8)	0.294
**Ultrasound characteristics**				
Solitary nodule, n (%) (multi)nodular goiter with dominant nodule, n (%) (multi)nodular goiter without dominant nodule, n (%) Cyst, n (%) No nodular lesions, n (%)	76 (39.0) 55 (28.2) 49 (25.1) 7 (3.6) 8 (4.1)	83 (23.6) 142 (40.5) 66 (18.8) 39 (11.1) 20 (5.7)	159 (29.1) 197 (36.1) 115 (21.1) 46 (8.4) 28 (5.1)	<0.001 0.004 0.085 0.002 0.414
Maximum diameter nodule				
<1 cm, *n* (%)	29 (14.9)	14 (4.0)	43 (7.9)	<0.001
1–2 cm, *n* (%)	66 (33.8)	68 (19.4)	134 (24.5)	0.001
2–4 cm, *n* (%)	71 (36.4)	167 (47.6)	238 (43.6)	0.001
>4 cm, *n* (%)	11 (5.6)	49 (14.0)	60 (11.0)	0.001
**Patients with FNAC, n (%)** **Number of nodules undergoing FNAC, n**	122 (62.6) 128	245 (69.8) 257	367 (67.2) 385	0.084

^#^ *p*-value for the comparison between incidentalomas and non-incidentalomas; * data are missing for 31 patients.

**Table 2 cancers-15-02350-t002:** Clinical characteristics of patients with malignant tumors. TC: thyroid carcinoma.

	Incidentaloma	Non-Incidentaloma
**Malignant**	10	39
Papillary TC	8 (80%)	23 (59%)
Follicular TC	1 (10%)	5 (12.8%)
Medullary TC	0	4 (10.3%)
Anapalastic TC	0	5 (12.8%)
Metastasis of primary non-thyroid malignancy	1 (10%)	1 (2.6%)
Squamous cell carcinoma	0	1 (2.6%)
**ATA risk of non-medullary TC** [1]		
Low	3 (30%)	19 (49%)
Intermediate	4 (40%)	6 (15.5%)
High	1 (10%)	8 (21%)
Not applicable	1 (10%)	6 (15.5%)
**Response to therapy**		
Excellent response	7 (70%)	26 (67%)
Incomplete biochemical response	0	2 (5%)
Indeterminate	0	1 (2.5%)
Structural disease	1 (10%)	9 (23%)
Not applicable	2 (20%)	1 (2.5%)

## Data Availability

The data presented in this study are available on request from the corresponding author.

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
