# Peer review of "Clinical Characteristics, Diagnostic Approach and Outcome of Thyroid Incidental Findings vs. Clinically Overt Thyroid Nodules: An Observational Single-Centre Study"

_cancers, 2023, doi:10.3390/cancers15082350_

Round 1

Reviewer 1 Report

This is an interesting study on the comparison of incidentalomas vs non-incidentalomas in the thyroid. I think it will be a useful addition to the already existing literature. However, I have several comments:

1) While English is of high quality, the manuscript is very hard to follow. It takes a considerable amount of time to go over the results section. I think this section needs to be re-written. The authors should remove unnecessary information, and should not duplicate what they are showing in the tables. For example lines 164-208 can be significantly reduced in length, as it's all info that can be found on Table 1.

2) I would suggest to add a Figure that shows the algorithm and the numbers of nodules, as this can be a great way to schematically represent the cohort and what patients were excluded at each point along the process. That will make it easier for the reader to visualize and follow the manuscript.

3)It is imperative that the authors go over the results can have in every percentage they use a numerator and a denominator. For example 9% (9/100). They should stay consistent throughout the manuscript as otherwise it is almost impossible to compare percentages and understand if that is an appropriate comparison.

4)The authors are comparing every little detail, and that way there is no cohesion to the message they are trying to get across. I would recommend removing unnecessary details, and trying so simplify message. That way an, otherwise, interesting message gets diluted.

5)The malignancy rate between the incidentalomas and non-incidentalomas was the same between the two groups when the authors looked at the surgeries that were recommended. Then when they looked at overall surgeries, the percentage of cancers in non-incidentalomas. How do the authors explain these findings?  I am not sure I can find in the paper if there is a difference between the "surgeries recommended" and the "surgeries performed" (if there is any). Is this because to the higher number of surgeries? If so, what are the tumors that were found in the percent of people that were found to have cancer but did not have a "surgery recommended"? Were they small incidental PTCs, and therefore, not clinically significant? That would probably be the most interesting part of the paper, and would convey the message, that in incidentalomas or non, thyroid US continues to be the best modality in exploring chances of malignancy and therefore surgeries should not be performed unless indicated  

Author Response

  1. While English is of high quality, the manuscript is very hard to follow. It takes a considerable amount of time to go over the results section. I think this section needs to be re-written. The authors should remove unnecessary information, and should not duplicate what they are showing in the tables. For example lines 164-208 can be significantly reduced in length, as it's all info that can be found on Table 1.

Answer: We thank the reviewer for the valuable comments and for the positive assessment of our paper. As recommended, we have largely rewritten the results section and we have removed redundant information.

  1. I would suggest to add a Figure that shows the algorithm and the numbers of nodules, as this can be a great way to schematically represent the cohort and what patients were excluded at each point along the process. That will make it easier for the reader to visualize and follow the manuscript.

Answer: As suggested, we have added a figure (Figure 1 in the revised manuscript) illustrating the flow of patients so that the numbers are easier to follow and to better visualize the results.

  1. It is imperative that the authors go over the results can have in every percentage they use a numerator and a denominator. For example 9% (9/100). They should stay consistent throughout the manuscript as otherwise it is almost impossible to compare percentages and understand if that is an appropriate comparison.

Answer: as recommended, to improve the readability of the paper we went over the numbers and percentages and presented them uniformly across the whole paper as percentage followed by number affected/total number within the brackets. 

  1. The authors are comparing every little detail, and that way there is no cohesion to the message they are trying to get across. I would recommend removing unnecessary details, and trying so simplify message. That way an, otherwise, interesting message gets diluted.

Answer: we thank the reviewer for the suggestion. As recommended we have removed redundant details to simplify the message. 

  1. The malignancy rate between the incidentalomas and non-incidentalomas was the same between the two groups when the authors looked at the surgeries that were recommended. Then when they looked at overall surgeries, the percentage of cancers in non-incidentalomas. How do the authors explain these findings?  I am not sure I can find in the paper if there is a difference between the "surgeries recommended" and the "surgeries performed" (if there is any). Is this because to the higher number of surgeries? If so, what are the tumors that were found in the percent of people that were found to have cancer but did not have a "surgery recommended"? Were they small incidental PTCs, and therefore, not clinically significant? That would probably be the most interesting part of the paper, and would convey the message, that in incidentalomas or non, thyroid US continues to be the best modality in exploring chances of malignancy and therefore surgeries should not be performed unless indicated  

Answer: We thank the reviewer for giving us the possibility to clarify this issue. To help visualize the patients flow we have also provided a new Figure 1 in the revised manuscript. The malignancy rate in the overall group (i.e.: for example also including patients in whom follow-up was not indicated at all according to the Dutch protocol) was higher in the non-incidentalomas than in incidentalomas. This is likely because based on clinical and sonographic aspects and FNAC results malignancies were less frequently suspected and hence surgery was significantly less frequently recommended and performed in the incidentaloma group, compared to the non-incidentaloma group (15.4 vs 29.6%). One could argue that malignancies might have been missed in the incidentaloma group, because less surgeries were performed. However: because there was no significant difference in the malignancy rate between incidentalomas and non-incidentalomas who had surgery recommended, we think this is less likely. Concluding, we believe that the overall difference is likely to be caused by the fact that non-PET incidentaloma group contains less malignancies than non-incidentaloma group. The similarity between the groups who had surgery is likely because the indication was correctly assessed.

In total, 134 patients had an indication/recommendation for surgery. Of these, in 6 patients who had an indication/recommendation for surgical removal, surgery was not performed or data are missing. This is due to: not undergoing surgery because of inoperable tumor or comorbidity (n=3), lost to follow-up (n=2), surgery took place in a different hospital and data retrieval was not successful (n=1). This is described in the results section (lines 255-259). The 5 patients in which no surgery was performed but had malignancy proven by cytology, the malignancies included: 1 papillary TC, 2 medullary TC, 1 anaplastic TC and 1 metastasis from renal cell carcinoma.

There are no patients who had no surgery recommended but had removal of the thyroid (lobe) anyway.

In the revised manuscript we have mentioned these aspects in lines 255-259 and 291-294

We also agree with the reviewer that the US features can help refine the selection of nodules who require immediate attention. We have added a mention to this aspect in the conclusion of the paper (lines 421-423) and in the abstract.

Reviewer 2 Report

In this study, the authors address an interesting topic, which actually still has unanswered questions: how to treat thyroid nodules found incidentally. However, the retrospective nature, the limited series and other limitations that I will list, make this study also inconclusive on the issue.

The results are pretty obvious too, not surprising. They confirm what was already known from other retrospective series.

Major comments:

-line 146: "follow-up at our clinic or at the referring center else- 146 where when patients were referred form another center". How could they rigorously and uniformly collect follow-up data from patients being followed up at other centers? They say it's a single-center study.

- line 152: according to their definition, therefore, nodules found in subjects with, for example, hyperthyroidism or Hashimoto's thyroiditis would also be incidentalomas. If we consider only palpable or symptomatic nodules as "non-incidentalomas", I would expect less than 64%. I also suggest better defining in the methods what is meant by "symptomatic", as the authors do in table 1.

- line 178: what symptoms? compressive?

- Table1: a very short median follow-up emerges (2.9 years). Too short for very slow growing lesions such as thyroid nodules, even when malignant. I believe that, given that this data was collected up to 2018 and 5 more years have passed, the cohort needs to be improved and updated.

- line 234: also consider lost to follow-up? Especially if they have been entrusted to other centres, how do they get this information?

- line 237: Were the US repeated by the same operators with the same machines? Otherwise this data loses all meaning.

- Table 2: spell out acronyms. Does ATC mean anaplastic thyroid cancer? If yes, would that mean that 13% of non-accidents were anaplastic? Truly surprising, if not far-fetched.

Then I suggest to better explain the data "metastasis". Is it N (lymph node metastases) or M (distant)?

Author Response

  1. line 146: "follow-up at our clinic or at the referring center else- 146 where when patients were referred form another center". How could they rigorously and uniformly collect follow-up data from patients being followed up at other centers? They say it's a single-center study.

Answer: we thank the reviewer for giving us the opportunity to clarify this. As explained in the revised manuscript, after the initial assessment at the outpatient clinic (with or without FNAC), different approaches were recommended. When follow-up was recommended, the decision to perform this in our center or by the referring specialist (or family physician) was at liberty of our specialist and considered also the most patient friendly option (travel distance, as some of the patients were traveling from different cities). In 10.9% of the patients who were recommended follow-up, this was performed by the referring specialist of family physician. We advised them to refer the patient back if mechanical complaints would arise or when the nodule was believed to have increased in size. To the best of our knowledge, this has not occurred since data collection. This has been clarified in the revised manuscript (lines 401-406). Additionally, we refer to the answers to questions 4 and 5 for further clarification (if needed).

  1. line 152: according to their definition, therefore, nodules found in subjects with, for example, hyperthyroidism or Hashimoto's thyroiditis would also be incidentalomas. If we consider only palpable or symptomatic nodules as "non-incidentalomas", I would expect less than 64%. I also suggest better defining in the methods what is meant by "symptomatic", as the authors do in table 1.

Answer: The definition for incidentaloma, as defined in the national Dutch guidelines, was a thyroid nodule(s) incidentally found when imagine studies are performed for other indications than the thyroid gland. In all patients referred for a thyroid nodule thyroid function tests are being performed after the discovery of the nodule. Therefore, if a patient underwent for example a CT angiogram because of cardiac symptoms on which a thyroid nodule was found incidentally, this would be considered an incidentaloma even in the presence of concomitant thyroid dysfunction, which was assessed after the nodule was found. The patients who were referred for thyroid dysfunction and underwent additional imaging procedures were not included in the present series, as they represent a population with a different presentation. Indeed, we found a small percentage of thyroid dysfunction in the incidentaloma group, and the incidentaloma group contained significantly less (mostly subclininical) hypothyroid patients. We have clarified this point in lines 196-199.

  1. line 178: what symptoms? compressive?

Answer: As suggested, we have clarified the type of symptoms in Table 1. 

  1. Table1: a very short median follow-up emerges (2.9 years). Too short for very slow growing lesions such as thyroid nodules, even when malignant. I believe that, given that this data was collected up to 2018 and 5 more years have passed, the cohort needs to be improved and updated.

Answer: We understand the reviewer’s concern as thyroid nodules are quite slowly growing. Nonetheless, we would like to point out that historically, in the Netherlands the emphasis of diagnosis and treatment is put on clinically relevant thyroid nodules and therefore it is not recommended to routinely follow-up the patients with thyroid nodules, if they are asymptomatic. Moreover, it is left to the liberty of the specialist if he/she considers it necessary to follow up the nodules for a short period of time to exclude the fast-growing nodules that might require therapy or when the patients cannot be sufficiently reassured and specifically request repeating the US (which is rarely the case). Therefore, the large majority of the patients do not remain under the follow-up of a specialist, are referred to the family physician who is advised to refer the patient back to the specialist when there are mechanical or cosmetic issues or where the nodule is believed to have increased in size. This practice is believed to result in a lower number of clinically occult thyroid cancers in the Netherlands, than globally reported, and has not resulted in a lower survival rate in the Netherlands, as reported by the epidemiological data from the Dutch National Cancer Registry (www.iknl.nl/nkr). However, not surprisingly it has resulted in a very low number of thyroid microcarcinoma. To date, we can confirm that to the best of our knowledge, since the data has been collected, there have been no patients in whom a malignant tumor has been diagnosed. We cannot assess objectively (by US) whether there are nodules that might have slightly increased in size after the last US follow-up. We have clarified these points in the revised manuscript (lines 401-406).

  1. - line 234: also consider lost to follow-up? Especially if they have been entrusted to other centres, how do they get this information?

Answer: We thank the reviewer for giving us the chance to clarify this point. Reiterating the point that in the Netherlands the routine follow-up for patients with thyroid nodules is historically very restrictive, for the patients referred from the centers affiliated to our center, we indeed referred some patients back to the centers where they were initially referred from. When during the follow-up a thyroid cancer is being diagnosed, those patients are all discussed in our multidisciplinary regional tumor board which is chaired by our center. Although we cannot provide the US follow-up data from those patients, we can confirm that to date, to the best of our knowledge there were no additional patients with a malignant thyroid tumor discovered after the follow-up at our clinic has ended. We have clarified this issue in the revised manuscript (lines 401-406).

- line 237: Were the US repeated by the same operators with the same machines? Otherwise this data loses all meaning.

Answer: We can confirm that the follow-up US was repeated by the same operators using the same machine.

  1. - Table 2: spell out acronyms. Does ATC mean anaplastic thyroid cancer? If yes, would that mean that 13% of non-accidents were anaplastic? Truly surprising, if not far-fetched.

Answer: As requested we have spelled out the acronyms in Table 2. We confirm that the ATC was anaplastic thyroid carcinoma. The relatively high percentage of ATC is not surprising in our hands, given the fact that our center is a referral center of thyroid carcinoma in the Netherlands and we receive a relatively high number of patients with ATC. This has been clarified in the revised manuscript (lines 289-291 )

  1. Then I suggest to better explain the data "metastasis". Is it N (lymph node metastases) or M (distant)?

Answer: As suggested, we have explained the definition of metastasis better. There was no metastasized thyroid cancer but exclusively metastases of other primary tumors to the thyroid. (see Table 2)

Reviewer 3 Report

Tom Jansen et al. carried out a retrospective study comparing clinical characteristics, diagnostic approaches and clinical outcome between patients referred with thyroid incidentalomas and non-incidentalomas to one academic thyroid clinic. 351 patients (64.3%) were referred for non-incidentalomas and 195 (35.7%) for incidentalomas. They conclude that the study supports the current recommendations from the Dutch guidelines to not indiscriminately perform additional analysis and treatment on all incidentalomas, but prioritize this to FDG-PET-incidentalomas and clinically relevant non-PET-incidentalomas.

The study tries to address a still not completely resolved issue: the selection of thyroid nodules that need cytological investigation and, possibly, surgical treatment. This challenge is particularly important regarding incidentalomas. However, The current study presents several weaknesses and the conclusions are partially not supported by the data. The most important weakness of the study is its retrospective design and the consequent lack of some data along with the possible presence of bias in analyzing and interpreting the data.

Specific points

lines#84-86: the authors state that "The risk for malignancy in incidentalomas is presumed to be lower than that of palpable nodules" however, a recent systematic review (Clin Endocrinol (Oxf). 2022;96(2):246-254. doi: 10.1111/cen.14575) concluded that "Current evidence suggests that investigation and management of thyroid nodules should not be influenced by the mode of detection." Accordingly - as correctly reported by the authors- The most recent guideline of the American Thyroid Association does not distinguish between the approach towards incidentaloma vs. non-incidentalomanwith the exception of FDG-PET incidentaloma". Indeed, ATA guidelines suggest to select thyroid nodules for FNAC examination according to US features, family history of thyroid cancer, history of previous X ray therapy etc. In other words, the same criteria utilized by the authors to select nodule for FNAC examination.

The authors state that "Significantly less malignancies were found among incidentalomas compared to non-incidentalomas (5.1% vs. 11.1%, 50 P=0.019)" but do not underline that among nodules submitted to FNAC examination the overall prevalence nodules with Bethesda 3-6 was similar among incidentalomas and non -incidentalomas. This finding, in my opinion, confirms the ATA suggestions and the effectiveness of nodules selection according to the above reported characteristics regardless the mode of their selection.

Author Response

  1. Lines#84-86: the authors state that "The risk for malignancy in incidentalomas is presumed to be lower than that of palpable nodules" however, a recent systematic review (Clin Endocrinol (Oxf). 2022;96(2):246-254. doi: 10.1111/cen.14575) concluded that "Current evidence suggests that investigation and management of thyroid nodules should not be influenced by the mode of detection." Accordingly - as correctly reported by the authors- The most recent guideline of the American Thyroid Association does not distinguish between the approach towards incidentaloma vs. non-incidentalomanwith the exception of FDG-PET incidentaloma". Indeed, ATA guidelines suggest to select thyroid nodules for FNAC examination according to US features, family history of thyroid cancer, history of previous X ray therapy etc. In other words, the same criteria utilized by the authors to select nodule for FNAC examination.

The authors state that "Significantly less malignancies were found among incidentalomas compared to non-incidentalomas (5.1% vs. 11.1%, 50 P=0.019)" but do not underline that among nodules submitted to FNAC examination the overall prevalence nodules with Bethesda 3-6 was similar among incidentalomas and non -incidentalomas. This finding, in my opinion, confirms the ATA suggestions and the effectiveness of nodules selection according to the above reported characteristics regardless the mode of their selection.

Answer: we thank the reviewer for this comment and the opportunity for us to clarify these points. Indeed, the amount of FNAC Bethesda classification 3-6 is comparable between both groups, indeed likely due to the robust criteria for selection as also indicated by the reviewer. The majority of these patients had surgical removal, which resulted in the definitive histological diagnosis which are presented in the manuscript and showed significantly more malignancies in non-incidentalomas than incidentalomas. Those patients who did not receive surgery, were followed-up. As is showed in lines 226-233 of the revised manuscript, the vast majority of nodules remained stable during follow-up. Theoretically, it is possible that in this group of patients with Bethesda 3-6 who did not receive surgery, some malignancies were missed. However, if this was the case it is likely they were low-risk TC’s and in the absence of high-risk features, these tumours have an excellent outcome even if the surgery was delayed. This is in line with the current shift towards watchful waiting and a less aggressive initial treatment in patients with low risk patients as reported repeatedly in the literature. Therefore, we believe our findings support the recommendations by the Dutch guidelines to not indiscriminately perform additional analysis on non-PET incidentalomas. This practice is believed to have resulted in a lower number of clinically occult thyroid cancers in the Netherlands, than globally reported, and has not resulted in a lower survival rate in the Netherlands, as reported by the epidemiological data from the Dutch National Cancer Registry (www.iknl.nl/nkr). However, not surprisingly it has resulted in a very low number of thyroid microcarcinoma.

Reviewer 4 Report

This paper focuses on the diagnostic approccio and clinica outcome of incidentally discutere vs clinically evident thyroid nodules. The issue appear of huge interest in nowadays clinical contest.

The test is Wellington written and easy to understand.

However I suggest to revise figure 2 for clarity: the exploit pie-chart related to malignancy reporter perchantages which are related to the whole population.

Moreover, results section is too verbous.

Author Response

This paper focuses on the diagnostic approccio and clinica outcome of incidentally discutere vs clinically evident thyroid nodules. The issue appear of huge interest in nowadays clinical contest.

The test is Wellington written and easy to understand.

However I suggest to revise figure 2 for clarity: the exploit pie-chart related to malignancy reporter perchantages which are related to the whole population.

Moreover, results section is too verbous.

Answer: We thank the reviewer for the positive assessment of our paper. As suggested we have provided the malignancy percentages for the whole population in Figure 1.  Moreover, we have rewritten the results sections to avoid redundancy and improve the readability of the paper.

Round 2

Reviewer 1 Report

The authors have successfully addressed my comments.

Author Response

We thank you for your constructive feedback and the time you have spent on the assessment of our paper. 

Reviewer 2 Report

The authors answered extensively to my comments and I think the manuscript is now much improved.

Author Response

(The authors gave the same response as above.)

Reviewer 3 Report

The authors partially addressed the questions and comments I  have previously raised. Specifically, recommendations of the American Thyroid Association for  selecting thyroid nodules to receive FNAC examination, which do not distinguish between incidentalomas and non-incidentalomas, in my opinion still remain useful.

Author Response

We thank you for your constructive feedback and the time you have spent on assessing our paper. 

In response to the comment we have added a specific note in the conclusion stating that the assessment of the nodules in our series resulted in similar number of FNAC, regardless the detection route and we have added a 
sentence in the conclusion conforming the usefulness of the ATA recommendations to assess the nodules and the indication for FNAC according to the sonographic characteristics.